# A Simple and Efficient Method for Designing Broadband Terahertz Absorber Based on Singular Graphene Metasurface

**DOI:** 10.3390/nano9101351

**Published:** 2019-09-20

**Authors:** Zhongmin Liu, Liang Guo, Qingmao Zhang

**Affiliations:** Guangzhou Key Laboratory for Special Fiber Photonic Devices and Applications & Guangdong Provincial Key Laboratory of Nanophotonic Functional Materials and Devices, South China Normal University, Guangzhou 510006, China; Liuzm@m.scnu.edu.cn (Z.L.); guoliangchn@163.com (L.G.)

**Keywords:** broadband absober, graphene metasurface, circuit model

## Abstract

In this paper, we propose a simple and efficient method for designing a broadband terahertz (THz) absorber based on singular graphene patches metasurface and metal-backed dielectric layer. An accurate circuit model of graphene patches is used for obtaining analytical expressions for the input impedance of the proposed absorber. The input impedance is designed to be closely matched to the free space in a wide frequency range. Numerical simulation and analytical circuit model results consistently show that graphene metasurface-based THz absorber with an absorption value above 90% in a relative bandwidth of 100% has been achieved.

## 1. Introduction

Terahertz (0.1–10 THz) absorbers have attracted great interest for their extensive applications in sensing [1], imaging [2], detecting [3], modulation [4], thermal emitters [5], and so on. However, THz electromagnetic waves are very hard to be detected by natural materials [6]. Metamaterial absorbers (MAs), a new type of artificially structured electromagnetic materials, have drawn much attention in recent years due to its extraordinary electronic and optical properties [7,8,9,10,11]. This is especially important for a broad THz frequency range that can realize near-unity absorption [12,13].

Graphene, a material consisting of one monolayer of carbon atoms, has been widely investigated as a promising platform for designing broadband tunable MAs in THz regime due to its exotic properties, such as optical transparency, high absorption, high electron mobility [14,15]. In addition, its sheet conductivity depending on chemical potential can be continuously tuned in a broad frequency range by tuning the gate voltage or chemical doping [16,17]. Recently, several broadband graphene-based MAs have been proposed [18,19,20,21,22,23,24]. However, most of these works only show specific structures having broadband absorption and few investigations explain the way to design the structure in order to get the desired broadband absorption in the graphene-based MAs.

In this paper, a simple and efficient method for designing broadband THz MAs based on a periodical monolayered graphene patch array has been proposed. This method is based on the equivalent circuit model, in which an analytical surface admittance is employed to describe the properties of the graphene metasurface. Moreover, Conditions for broadband absorption have been analyzed. A set of closed-form relations for material properties and geometrical parameters of the absorber have been derived. As a consequence, we can successfully design a broadband THz absorber for a given central frequency based on the formulas derived from this approach. To validate the approach, the results have been compared with those obtained by the finite-difference time-domain (FDTD Solutions, Lumerical Inc., Vancouver, BC, Canada) simulations, excellent agreements have been achieved.

This paper is organised as follows. The equivalent circuit model of the proposed device is derived in Section 2. The design scheme leading to an broadband absorber is presented in Section 3. Finally, conclusions are given in Section 4.

## 2. Equivalent Circuit Model Approach

To realize electromagnetic wave absorbers at THz frequency, we can take a simple structure. A schematic of the proposed absorber is shown in Figure 1. The structure consists of a sheet of periodic graphene patches on the top, a dielectric spacer layer, and a ground plate on the bottom. The graphene patches are periodically arranged along the *x* and *y* directions and embedded into the dielectric material. This layer is used to induce high absorption. The dielectric spacer layer has a refractive index of $n_{d}=1.5$ and low losses in THz band. The bottom layer is made of gold with a conductivity of $4\times 10^{7}$ S/m and a thickness of 0.5 μm. As the thickness of the gold used here is much lager than the typical skin depth in the THz regime, the gold layer is acting as a reflector.

As we know, the surface conductivity of graphene is composed of interband and intraband contributions derived using the Kubo formula [25]. In the THz regime, where the photon energy $\hbar \omega \ll 2\mu_{c}$ ($\mu_{c}$ is the chemical potential), the interband contribution of the surface conductivity can be neglected. Thus, the surface conductivity of the graphene has a Drude form [26,27]:(1)$$\sigma_{g}=\frac{\sigma_{0}}{1+j\omega \tau }$$
where
(2)$$\sigma_{0}=\frac{e^{2}k_{B}T\tau }{\pi \hbar_{2}}\left\{\frac{\mu_{c}}{k_{B}T}+2ln\left[exp\left(-\frac{\mu_{c}}{k_{B}T}\right)+1\right]\right\}$$
where *e* is the electron charge, $k_{B}$ is Boltzmann’s constant, *ℏ* is the reduced Planck’s constant, τ is the carrier relaxation time, *T* is the temperature, ω is the angular frequency. In this paper, we use T=300 K and $\tau =10^{-13}$ s [28].

Here, we use an equivalent circuit model to design the broadband graphene-based absorber. Based on the transmission line theory, the whole structure of the absorber can be considered as circuit elements [28]. The equivalent circuit of the device at normal incidence for TM polarization (electric field along the *x* direction) is illustrated in Figure 2. In this circuit model, the top graphene patch array can be modeled with a shunt admittance $Y_{g}$. The homogeneous regions, namely, the free space and the dielectric spacer are modeled as transmission lines. The bottom metal can be represented by the admittance $Y_{Au}$.

Models for the graphene patch array have been presented in several studies [29,30,31]. Barzegar-Parizi et al. proposed an accurate analytical R-L-C circuit model [29]. The results calculated by the circuit model agreed well with those by full-wave simulations, demonstrating that the circuit model is accurate and effective. However, the value of C in the formula has to be carried out numerically. Meanwhile, a simple closed-form expression for graphene patch array was introduced by Padooru et al. in Reference [30]. The model showed considerable discrepancies with the full-wave simulations, which was demonstrated in Reference [29]. Moreover, the capacitive impedance of metal square patches has been modeled by Luukkonen et al. in Reference [31]. Results by the models were verified with good accuracy compared with those by full-wave simulations. Combining the models proposed in References [29,31], here we propose a simple and accurate analytical circuit model for graphene patches. The surface admittance of the graphene patch array can be written as:(3)$$Y_{g}=1/\left[R+j\omega L+\frac{1}{j\omega C}\right]$$
(4)$$R=\frac{aD^{2}}{{\left(D-g\right)}^{2}}\cdot Re\left\{\sigma_{g}^{-1}\right\}$$
(5)$$L=\frac{aD^{2}}{{\left(D-g\right)}^{2}}\cdot \frac{Im\{\sigma_{g}^{-1}\}}{\omega }$$
(6)$$C=\frac{2D\varepsilon_{eff}}{\pi }\cdot ln\left\{csc\left(\frac{\pi g}{2D}\right)\right\}$$
where *a* is the weighted coefficient, *D* is the period of the patch array, *g* is the gap between the graphene patches (D>g), $\varepsilon_{eff}=\varepsilon_{0}\left(1+\varepsilon_{d}\right)/2$ is the average permittivity of the mediums surrounding the graphene patches, $\varepsilon_{0}$ is the dielectric constant of vacuum and $\varepsilon_{d}=n_{d}^{2}$ is the relative permittivity of the dielectric material. It can be seen that the graphene patch array can be modeled by a series R-L-C circuit, where *R* and *L* are produced by the graphene patches, corresponded to the surface impedance $1/\sigma_{g}$ and the geometric factor $aD^{2}/{\left(D-g\right)}^{2}$. we get a=1.13 by analyzing the fundamental resonant mode of the array. it should be noted that neglecting the effect of higher order modes of the patch array for simplicity of calculation does not affect the accuracy of the model [29]. The *C* is induced by the gaps between the patches, associated with the patch geometry and background environment [31].

The ground plate of Au can be represented by a short circuit which is suppressing the transmittance, that is $Y_{Au}\to \infty$. So the input admittance of the metal-backed dielectric layer can be expressed as:(7)$$Y_{c}=\frac{Y_{d}}{j\cdot tan(\beta d)}$$
where $Y_{d}=n_{d}/Z_{0}$ and $\beta =\omega n_{d}/c$ are the admittance and the propagation of the transmission line corresponding to the dielectric spacer layer, respectively. $Z_{0}=120\pi$, where $Z_{0}$ is the admittance of free space, *d* is the thickness of the dielectric, *c* is the speed of light in free space.

With the aid of the above analysis, the total input impedance of the absorber is
(8)$$Y_{in}=Y_{g}+Y_{c}$$

The reflection coefficient Γ from the top interface of absorber and free space is assessed as follows:(9)$$\Gamma =\frac{Z_{in}-Z_{0}}{Z_{in}+Z_{0}}$$

So the power absorbance *A* of the absorber is
(10)$$A=1-|\Gamma {|}^{2}$$

## 3. Broadband Absorber Design

We now design an THz absorber to achieve it broadband absorption. Firstly, we’d like to analyze the situation of achieving perfect absorption. It is well known that the conditions for perfect absorption are $Re\left(Y_{in}\right)=1/Z_{0}$ and $Im(Y_{in})=0$. It can be realized for a given central frequency, $\omega_{0}$, by setting $Re\left(Y_{g}\right)=1/Z_{0}$, $Im(Y_{g})=0$ and $Im(Y_{c})=0$. So the conditions for perfect absorption can be summarized as follow:(11)$$R=Z_{0}$$
(12)$$LC=\frac{1}{\omega_{0}^{2}}$$
(13)$$\beta d=\frac{\pi }{2}$$

Let us design a perfect absorber for a central frequency $\omega_{0}$ at 1 THz. The geometrical parameters of the designed graphene metasurface is assumed to be g/D=0.1, the period and gap of the graphene patches are given by Equations (Equation 11) and (Equation 12) as D=39.5
μm and g=4.0
μm, the chemical potential of the graphene patch array can be calculated from Equation (Equation 11) as $\mu_{c}=0.314$ eV. Finally, from Equation (Equation 13), we obtain the thickness of the dielectric spacer d=50
μm.

With the aid of the above analytical equations, a perfect absorber for a given central frequency is designed. Figure 3a shows the absorption of this structure calculated by the equivalent circuit model (dotted line) and FDTD simulations (solid line). An excellent agreement can be observed between the two results, which demonstrates that the equivalent circuit model can accurately reflect the absorber’s performance. It should be pointed out that there are slight disagreements around the second absorption peak due to the ignorance of higher order modes in the equivalent circuit model. Moreover, we compare the results obtained by the proposed circuit model with those obtained by the model of Reference [29]. The circles in Figure 3a show the results produced by the model of Reference [29]. The absorptions predicted by the two circuit models are almost coincide. This shows that it is effective to use a simple closed-form expression of C reported in Reference [31] instead of that proposed in [29].

It can be seen that the absorption decreases rapidly with the corresponding frequency away from the perfect matching point in Figure 3a. Moreover, the bandwith of 90% absorption is just about 0.73 THz. So the conditions for perfect absorption do not lead to broadband absorption. Furthermore, the real and imaginary parts of the input admittance of the designed absorber are depicted in Figure 3b. It can be seen that the sharp absorption spectrum of the perfect absorber is mainly caused by a steep change from 0 to large in the imaginary part of the equivalent circuit, whereas the changes around the central frequency in the real part are relatively flat.

To further verify the inpact of real and imaginary part of the input admittance upon the absorption of the absorber, We test the absorption spectra of a perfect absorber with the central frequency at 1.5 THz (Figure 4a). The bandwidth of 90% absorption reaches about 0.99 THz, which is much wider than the bandwidth at the central frequency of 1 THz (0.73 THz in Figure 3a). The real and imaginary part are also plotted in Figure 4b, which allows us to compare the trends around the central frequency of the two parts. A relatively gentle change in the imaginary part of the absorber with the central frequency of 1.5 THz is corresponding to a wider high absorption bandwidth around the central frequency. Therefore, in order to obtain broadband absorption, we attach an additional condition on the imaginary part of the input admittance:(14)$$\frac{d}{d\omega }Im\left(Y_{in}\right){|}_{\omega =\omega_{0}}=0$$

The additional condition leads to a small and slowly varying imaginary part in a wide range of frequencies near the central frequency [32]. With βd=π/2, the input admittance of the metal-backed dielectric layer around can be expressed by Taylor formula:(15)$$Im\left(Y_{c}\right){|}_{\omega =\omega_{0}}≅\frac{\pi n_{d}}{2\omega_{0}Z_{0}}\left(\omega -\omega_{0}\right)$$

Similarly, the admittance of the graphene metasurface for frequencies around $\omega_{0}$ can be expressed as:(16)$$Im\left(Y_{g}\right){|}_{\omega =\omega_{0}}≅-\frac{2L}{R^{2}}\left(\omega -\omega_{0}\right)$$

Differentiating Equations (Equation 15) and (Equation 16) at $\omega =\omega_{0}$, the resistance of the graphene array, *R*, is then obtained from Equation (Equation 14):(17)$$R=\frac{4\tau \omega_{0}Z_{0}}{\pi n_{d}}$$

Moreover, in order to fulfill the requirement of a broadband absorption (A>0.9), there is a limitation on the central frequency:(18)$${\left(\frac{R-Z_{0}}{R+Z_{0}}\right)}^{2}<0.1$$
which results in
(19)$$R>0.52Z_{0}$$

In conclusion. The conditions for broadband absorption can be summarised as Equations (Equation 12), (Equation 13), (Equation 17), and (Equation 19). Now let us design a broadband absorber. According to Equations (Equation 17) and (Equation 19), when the material has a dielectric constant at $n_{d}=1.5$, high absorption can be achieved at the central frequency of 1 THz. In the constraint condition (Equation (Equation 17)), we set the geometrical parameters of designed graphene metasurface at g/D=0.1 and get a chemical potential of the graphene patches at $\mu_{c}=0.589$ eV. The period and gap of patches can be easily calculated by Equation (Equation 12), one obtains D=74.1
μm and g=7.4
μm. Moreover, the thickness of the dielectric spacer is obtained from Equation (Equation 13) as d=50
μm.

The absorption of this structure is shown in Figure 5a. It can be observed that the effective absorption bandwidth (above 90%) is from 0.47 to 1.47 THz, indicating a relative absorption bandwidth of 100%. As it is shown that the real and imaginary parts of the input admittance, $Y_{in}$, in Figure 5b. The imaginary part of the input admittance is near zero and changes slowly around the central frequency, while the real part of the input admittance remains a small reduction near $Z_{0}$ in the corresponding frequency range. These features lead to wide-band impedance matching. Thus, a broadband absorption can be realized.

Considering the actual manufacturing errors, effect of the variations of Fermi energy and gap of graphene patches on the absorber performance are examined. As shown in Figure 6a, there are no significant change on high absorption bandwidth for ±5% variation in the Fermi energy of graphene patches. Moreover, it can be seen from Figure 6b that the high absorption characteristics of the absorber is not very sensitive for ±5% variation in the gap of graphene patches.

It should be pointed that the design algorithm has some limitations. Firstly, to achieve high absorption, *R* should be large than $0.52Z_{0}$. Since here τ is a constant, the central frequency cannot be chosen arbitrarily, That is $\omega_{0}/2\pi >0.65n_{d}$ THz. The other limitation deserves attention. The period of patch arrays must be much smaller than the wavelength and g/D must be much smaller than unity. Otherwise, the proposed circuit model would not be valid [33].

## 4. Conclusions

A novel approach for graphene metasurface-based broadband THz absorber design has been proposed. This approach significantly simplified the design of broadband THz absorbers. Based on the equivalent circuit model, the imaginary part of the absorber input admittance and its derivative are designed to be zero at a given center frequency. Then, the real part of the device input admittance is tuned to match the free-space admittance around the central frequency. As a result, a relative absorption bandwidth of 100% is realized by using only one layer of graphene metasurface. The results also have been compared with FDTD simulations and good agreements have been achieved. It can be concluded that the proposed approach can provide an effective method to analyze and design graphene-based broadband absorbers.

## Figures and Tables

**Figure 1 nanomaterials-09-01351-f001:**
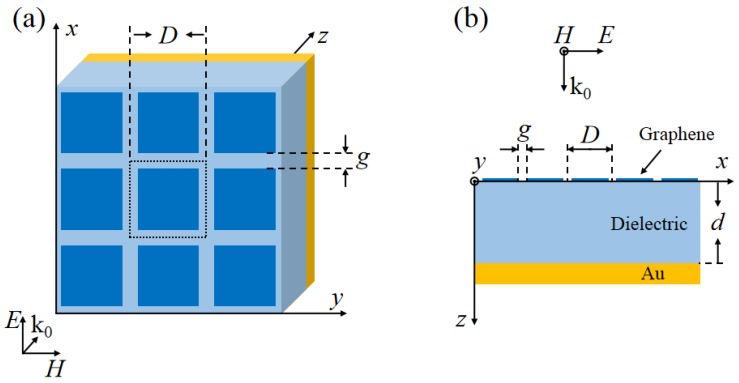
Schematic diagram of the proposed absorber: (**a**) perspective view. (**b**) side view.

**Figure 2 nanomaterials-09-01351-f002:**
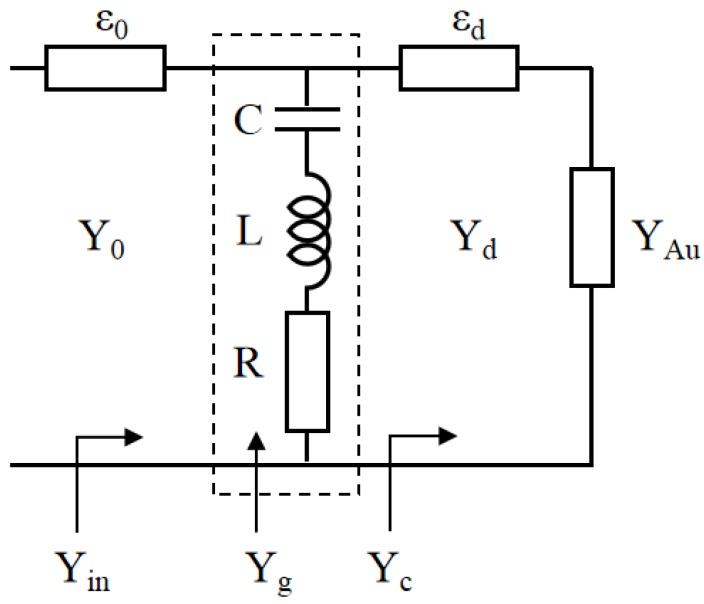
Equivalent circuit model for the graphene metasurface-based absorber.

**Figure 3 nanomaterials-09-01351-f003:**
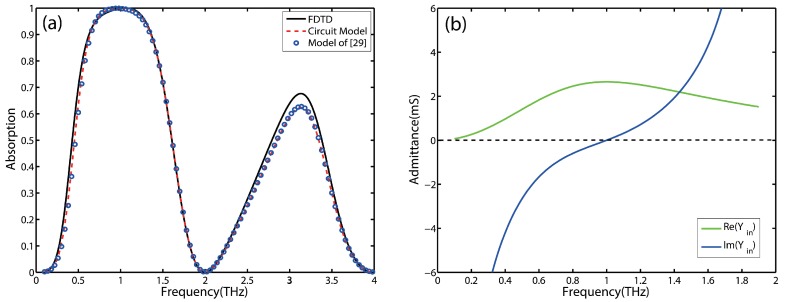
(**a**) The absorption spectrum of the designed perfect absorber with the central frequency at 1 THz simulated by FDTD simulations and calculated by circuit model. (**b**) Real and imaginary parts of the input admittance of the designed absorber.

**Figure 4 nanomaterials-09-01351-f004:**
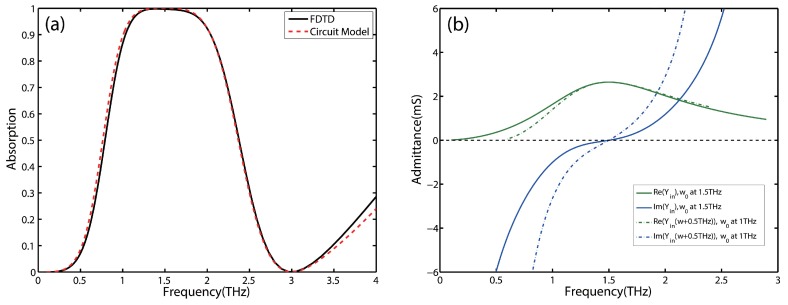
(**a**) The absorption spectrum of the designed perfect absorber with the central frequency at 1.5 THz simulated by FDTD simulations and calculated by circuit model. (**b**) The trends of real and imaginary of the input admittance of the designed absorber with the central frequency at 1.5 THz vs. that of the prefect absorber with the central frequency at 1 THz.

**Figure 5 nanomaterials-09-01351-f005:**
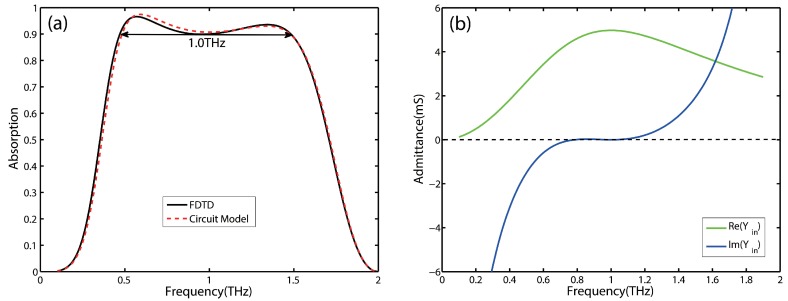
(**a**) The absorption spectrum of the designed broadband absorber with the central frequency at 1 THz simulated by FDTD simulations and calculated by circuit model. (**b**) Real and imaginary parts of the input admittance of the designed absorber.

**Figure 6 nanomaterials-09-01351-f006:**
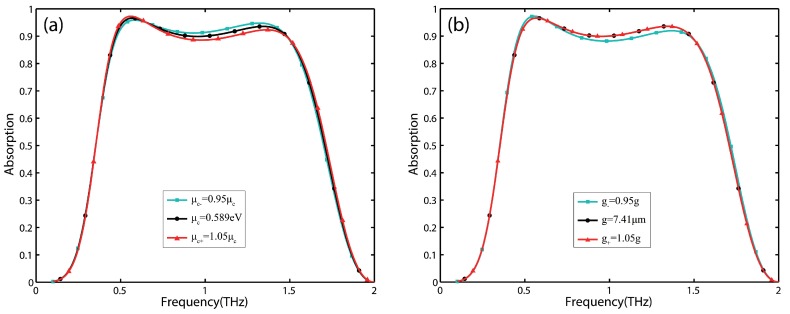
The absorption spectrum of the absorber calculated by full-wave simulation for: (**a**) three different Fermi energy of graphene patches. (**b**) three different gaps of graphene patches.

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
