# Peer review of "A Simple and Efficient Method for Designing Broadband Terahertz Absorber Based on Singular Graphene Metasurface"

_nanomaterials, 2019, doi:10.3390/nano9101351_

Round 1

Reviewer 1 Report

The authors proposed a simple and efficient method for designing a THz-broadband absorber based on graphene metasurface effects. An accurate circuit model of graphene patches is proposed additonialy for obtaining analytical expressions for the input impedance of the proposed absorber.  The analytical circuit model demonstrates that the graphene based THz absorber with an absorption value above 90% can be achieved.

To my opinion, the manuscript is well written and the results are of huge scientific interest.

I don't have any suggestions how to improve the quality of this paper.

Author Response

Summary and general comment: TThe authors proposed a simple and efficient method for designing a THz-broadband absorber based on graphene metasurface effects. An accurate circuit model of graphene patches is proposed additonialy for obtaining analytical expressions for the input impedance of the proposed absorber.  The analytical circuit model demonstrates that the graphene based THz absorber with an absorption value above 90% can be achieved.

Response : Thank you for the comments.

Point 1: To my opinion, the manuscript is well written and the results are of huge scientific interest.

I don't have any suggestions how to improve the quality of this paper.

Response 1: Thank you very much for your nice comments, we would like to express our greatest appreciation.

Reviewer 2 Report

Summary and general comment:

This research reports the use of a periodical monolayered graphene patch array for the design of broadband terahertz metamaterial absorbers based. The proposed method and analytical results are built on the equivalent circuit model approach. Subsequently, the conditions for broadband absorption have been analyzed for understanding the material properties and the geometrical parameters. In addition, in order to validate the approach, the experimental results have been compared with those obtained from full-wave simulation using FDTD solutions software. A few additional comments below are listed for the authors as a reference.

Additional Comments:

Based on the transmission line theory, the authors propose the structure of the absorber as circuit elements. However, the physical properties of each unit and their application as a device need more explanation and discussion.

The experimental results are compared with full-wall simulation, and display good agreement. However, the readers might also be interested in the misfit data. Could the authors explain more about this?

Although this research aims at proposing a calculation, experimental data collected from real sample would be very appreciated.

Some minor corrections are suggested in the following list. Please check the missing spaces around “,”
Line 4: … of the proposed “absorber.”
Line 81: …a given central “frequency, w0,” by …
Line 130: …of the input admittance, “Yin,” in Fig.5(b) …

Author Response

Summary and general comment: This research reports the use of a periodical monolayered graphene patch array for the design of broadband terahertz metamaterial absorbers based. The proposed method and analytical results are built on the equivalent circuit model approach. Subsequently, the conditions for broadband absorption have been analyzed for understanding the material properties and the geometrical parameters. In addition, in order to validate the approach, the experimental results have been compared with those obtained from full-wave simulation using FDTD solutions software. A few additional comments below are listed for the authors as a reference.

Response : Thank you for the comments.

Point 1: Based on the transmission line theory, the authors propose the structure of the absorber as circuit elements. However, the physical properties of each unit and their application as a device need more explanation and discussion.

Response 1: We have supplemented this part on page 2 according to the Reviewer’s suggestion. Namely: “The graphene patches are periodically arranged along the x and y directions and embedded into the dielectric material, this layer is used to induce the high absorption. The dielectric spacer layer has a refractive index of nd = 1.5 and low losses in THz band. The bottom layer is made of gold with a conductivity of 4×107 S/m and a thickness of 0.5 μm. As the thickness of the gold used here is much lager than the typical skin depth in the THz regime, the gold layer is acting as a reflector.” 

Point 2: The experimental results are compared with full-wall simulation, and display good agreement. However, the readers might also be interested in the misfit data. Could the authors explain more about this?

Response 2: Thank you for your suggestion, and we have revised the manuscript as follows on page 3: “The patch array is modeled by a shunt admittance Yg which represents an infinite number of parallel R-L-C circuits, each corresponding to one mode of graphene patches.” The equivalent circuit model in this paper is based on the fundamental resonant mode of graphene patches. There are slight disagreements between equivalent circuit model and FDTD simulations due to the ignorance of higher order modes in the equivalent circuit model.

Point 3: Although this research aims at proposing a calculation, experimental data collected from real sample would be very appreciated.

Response 3: We appreciate your suggestion on collecting experimental data from real sample. However, this article focuses on the method of designing broadband absorbers. We use numerical calculations rather than experimental verification, because they have the advantage of eliminating interference caused by real manufacturing measurement deviations. The experimental measurement would be our next step. Thank you for your good suggestion.

Point 4: Some minor corrections are suggested in the following list. Please check the missing spaces around “,”

Line 4: … of the proposed “absorber.”

Line 81: …a given central “frequency, w0,” by …

Line 130: …of the input admittance, “Yin,” in Fig.5(b) …

Response 4: We are very sorry for our negligence and have made correction on Page 1 line 4, Page 4 line 86, and Page 6 line 138.

Reviewer 3 Report

Review of the manuscript „A simple and efficient method for designing broadband terahertz absorber based on singular graphene metasurface“

The author report about a way to avoid computational time consuming full-wave simulations while designing patch array THz broadband absorbers made from graphene. The authors aim is to provide an analytic expression for calculation the dimensions of the structure and to achieve a combination of high absorption and broadband characteristic. This analytic expression should avoid the time and cost consuming calculation of the structure by a full wave simulation (FDTD).

I have the following comments and remarks:

(1) Page 3, line 62: in order to achieve a good agreement between you calculation and the FDTD results you have to choose the weighted coefficient a = 1.13. Since the authors of Ref. [29] – you cited – evaluate their fitting parameter a from FDTD, the question is, if one could always use you’re a in the presented analytical ansatz, or if FDTD calculation is necessary in any case to find the fitting value of a. If the last would be the case then your analytic model as replacement for a FDTD calculation would be highly questionable. How stable are your results with respect of the correct choice of coefficient a?

(2) Eq. (1) and (2): please also introduce j and µ_c below the equation. Same with n_d below Eq. (6).

(3) Page 3, line 72: Should it be “Y_d” instead of “Y_c”?

(4) How much of the patches have to be illuminated to reach the calculated absorbance, since the calculation – if I understand it correctly – starts from an infinite large patch area?

(5) Some “part” are missing after “imaginary” throughout the manuscript.

(6) Page 7, Fig. 6: Are the variation proposed are stochastically distributed over the patches, which would come close to a real structure which has a fluctuations around one value but not in the same direction for each individual pad or do the calculations in Fig. 6 propose only a systematic offset, i.e., all pads same size but larger/small then calculated? The last situation is a bit unlikely in real systems? How would a stochastic distribution of geometry fluctuations alter the performance of the absorber?

(7) Page 7, line 153: I assume “wall” should be “wave”.

Grammar/language:

(I) Please include blanks between number and unit (e.g.:”1 THz”).

(II) Usually the units are not printed in italic.

(III) Please check the structure of the sentences, otherwise some parts of the manuscript are hard to understand, e.g.: p. 1, line 22f; page 4, line: 79

(IV) Page 5, line 102: “absorption”

(V) Please check for missing blanks in the text, e.g. after commas, at the beginning of a sentence, before opening brackets (p. 1, lines 2 and 10) etc.

(VI) “FDTD” is used on page one, but the abbreviation is introduced much later in the manuscript.

Author Response

Summary and general comment: Review of the manuscript “A simple and efficient method for designing broadband terahertz absorber based on singular graphene metasurface”.

The author report about a way to avoid computational time consuming full-wave simulations while designing patch array THz broadband absorbers made from graphene. The authors aim is to provide an analytic expression for calculation the dimensions of the structure and to achieve a combination of high absorption and broadband characteristic. This analytic expression should avoid the time and cost consuming calculation of the structure by a full wave simulation (FDTD).

Response : Thank you for the comments.

Point 1: Page 3, line 62: in order to achieve a good agreement between you calculation and the FDTD results you have to choose the weighted coefficient a = 1.13. Since the authors of Ref. [29] – you cited – evaluate their fitting parameter a from FDTD, the question is, if one could always use you’re a in the presented analytical ansatz, or if FDTD calculation is necessary in any case to find the fitting value of a. If the last would be the case then your analytic model as replacement for a FDTD calculation would be highly questionable. How stable are your results with respect of the correct choice of coefficient a?

Response 1: The patch array is modeled by a shunt admittance Yg which represents an infinite number of parallel R-L-C circuits. Each circuit corresponds to one mode of graphene patches. The proposed surface admittance of graphene patch array is based on the analysis of the fundamental mode of the graphene patch array. So the weighted coefficient a = 1.13 is a constant, which is from the calculation of the fundamental mode of the graphene patch array. We have validated the model under different sets of variables g, D and µc. The results calculated by the analytic model show good agreement with those by full-wave simulations. We added an explanation on page 3 according to the reviewer’s comments. 

Point 2: Eq. (1) and (2): please also introduce j and µ_c below the equation. Same with n_d below Eq. (6). 

Response 2: Thank you for your suggestion. We have introduced nd in Page 2 line 42 and µc in Page 2 line 47. j is an imaginary unit. Considering the universality of the Drude formulas, we therefore didn’t introduce this quantity in the manuscript.

Point 3: Page 3, line 72: Should it be “Y_d” instead of “Y_c”?

Response 3: We are very sorry for our negligence and have made a correction on page 4, line 77.

Point 4: How much of the patches have to be illuminated to reach the calculated absorbance, since the calculation – if I understand it correctly – starts from an infinite large patch area?

Response 4: This is a very good question. Previous research has reported that when the patches are larger than 20×20 array, the performance is comparable with the spectrum from infinite large patch area. While the patches are less than 20×20 array, the boundary effect needs to be considered.

Point 5: Some “part” are missing after “imaginary” throughout the manuscript.

Response 5: Thanks for your comments. We have corrected the typo.

Point 6: Page 7, Fig. 6: Are the variation proposed are stochastically distributed over the patches, which would come close to a real structure which has a fluctuations around one value but not in the same direction for each individual pad or do the calculations in Fig. 6 propose only a systematic offset, i.e., all pads same size but larger/small then calculated? The last situation is a bit unlikely in real systems? How would a stochastic distribution of geometry fluctuations alter the performance of the absorber?

Response 6: Yes indeed the value we proposed in Fig. 6 is a systematic value. However, the systematic fluctuations caused by photolithography in actual production process is also one of the dominant factors which is the main reason we demonstrating in the paper. Meanwhile, ±5% stochastic variation of g has no significant influence on overall performance of the absorber which can be neglected. We therefore only present the systematic variation in this work.

Point 7: Page 7, line 153: I assume “wall” should be “wave”.

Response 7: We are sorry for our negligence and have made correction.

Point 8: Grammar/language:

(I) Please include blanks between number and unit (e.g.:”1 THz”).

(II) Usually the units are not printed in italic.

(III) Please check the structure of the sentences, otherwise some parts of the manuscript are hard to understand, e.g.: p. 1, line 22f; page 4, line: 79

(IV) Page 5, line 102: “absorption”

(V) Please check for missing blanks in the text, e.g. after commas, at the beginning of a sentence, before opening brackets (p. 1, lines 2 and 10) etc.

(VI) “FDTD” is used on page one, but the abbreviation is introduced much later in the manuscript.

Response 8: We are sorry for our negligence. We have tried our best to improve the manuscript and made correction in the manuscript. Revised portion are underlined in red.

Reviewer 4 Report

The manuscript reports on a theoretical proposal to design a broadband THz absorber based on a sheet of periodic graphene patches.

The approach is based on the earlier models [29, 31], which relate to “complex” or “inaccurate” in line 58. If they are “inaccurate”, how the present model can be “accurate”? It is not indicated, what is the real simplification of the present approach. As soon as the earlier model [29] is variational, how can the present work be “accurate” and what is the measure of its “accuracy”? What is the origin and precision of the coefficient a=1.13 in line (62)? These issues are not discussed at all. 

The presentation is poor. For example, there are grammatical mistakes in lines 46, 56, 123, 141. Starting with Fig. 4, “real” and “imaginary” are erroneously used without “part”.  In line, 49, the expression “time of electro-phonon” (?) is incomplete. The Finite-Difference Time-Domain approach is called “full-wave simulation” (?) (line 91) or “full-wall simulation” (?) (line 153) without a necessary explanation of the essence of the performed simulations: which equations are simulated? Without this discussion, the meaning of the comparisons in Figs. 3 to 6 remain vague.

The manuscript is not suitable for publication in Nanomaterials.

Round 2

Reviewer 4 Report

The authors have significantly improved the presentation in the revised manuscript carefully taking into account my criticisms. The new version, however, still needs some improvements.

When claiming: “Results by the models were verified with good (?) accuracy compared with those by full-wave simulations” (lines 68, 69) or “here we get a simple and accurate analytical circuit model” (lines 70, 71), it is necessary, to provide the quantitative measure of the accuracy of the present method in comparison with that of its predecessors [29] and [31]. In spite of a number of revisions, the language remains below the standard of Nanomaterials. The unfortunate misprints like “sapcer” instead of “spacer” in line 59 must be thoroughly corrected by the authors. The grammar requires a profound refinement by the Editorial Staff.

With these improvements, the manuscript will become suitable for publication in “Nanotechnology”.
